# Study of the Genetic Variants in *BRCA1/2* and Non-*BRCA* Genes in a Population-Based Cohort of 2155 Breast/Ovary Cancer Patients, Including 443 Triple-Negative Breast Cancer Patients, in Argentina

**DOI:** 10.3390/cancers13112711

**Published:** 2021-05-31

**Authors:** Angela R. Solano, Pablo G. Mele, Fernanda S. Jalil, Natalia C. Liria, Ernesto J. Podesta, Leandro G. Gutiérrez

**Affiliations:** 1Genotipificación, Departamento de Análisis Clínicos, Centro de Estudios Médicos e Investigaciones Clínicas, Ciudad Autónoma de Buenos Aires C1431FWO, Argentina; fjalil@cemic.edu.ar (F.S.J.); nliria@cemic.edu.ar (N.C.L.); lgutierrez@cemic.edu.ar (L.G.G.); 2Instituto de Investigaciones Biomédicas, Facultad de Medicina, Universidad de Buenos Aires/Consejo Nacional de Investigaciones, Científicas y Técnicas, Ciudad Autónoma de Buenos Aires C1121ABG, Argentina; pgmele@fmed.uba.ar (P.G.M.); ernestopodesta@yahoo.com.ar (E.J.P.); 3Departamento de Bioquímica Humana, Facultad de Medicina, Universidad de Buenos Aires, Ciudad Autónoma de Buenos Aires C1121ABG, Argentina

**Keywords:** triple-negative breast cancer, NGS of gene panels, genetic predisposition to breast cancer, germline genetic testing

## Abstract

**Simple Summary:**

Gene/s sequencing in hereditary breast/ovary cancer (HBOC) in routine diagnosis is challenged by the analysis of panels. The aim of this report is to describe a retrospective analysis of *BRCA1/2* and non-*BRCA* gene sequencing in patients with breast/ovary cancer (BOC), including triple-negative breast cancer (TNBC). TNBC is associated to *BRCA1/2* at a higher rate than the rest of the breast cancer types. The more prevalent pathogenic variants (PVs) in *BRCA1/2* genes do not rule out the importance to panels of genes, although they are certainly far from shedding light on the gap of the 85% predicted linkage association of BOC with *BRCA1/2* and are still not elucidated. This data is also of value in health programming for alerting risks in breast screening and knowledge of the regional spectrum of genetic variants.

**Abstract:**

Gene/s sequencing in hereditary breast/ovary cancer (HBOC) in routine diagnosis is challenged by the analysis of panels. We aim to report a retrospective analysis of *BRCA1/2* and non-*BRCA* gene sequencing in patients with breast/ovary cancer (BOC), including triple-negative breast cancer (TNBC), in our population. In total 2155 BOC patients (1900 analyzed in *BRCA1/2* and 255 by multigenic panels) gave 372 (17.2.6%) and 107 (24.1%) likely pathogenic/pathogenic variants (LPVs/PVs), including *BRCA* and non-*BRCA* genes, for the total and TNBC patients, respectively. When BOC was present in the same proband, a 51.3% rate was found for LPVs/PVs in *BRCA1/2*. Most of the LPVs/PVs in the panels were in *BRCA1/2*; non-*BRCA* gene LPVs/PVs were in *CDH1, CHEK2, CDKN2A, MUTYH, NBN, RAD51D*, and *TP53*. TNBC is associated with *BRCA1/2* at a higher rate than the rest of the breast cancer types. The more prevalent PVs in *BRCA1/2* genes (mostly in *BRCA1*) do not rule out the importance to panels of genes, although they are certainly far from shedding light on the gap of the 85% predicted linkage association of BOC with *BRCA1/2* and are still not elucidated.

## 1. Introduction

Worldwide, breast cancer (BC) is the most common cancer in women, with diagnostics at a young age (pre-menopausal) being highly associated with hereditary factors increasing the risk of developing cancer [1]. Predisposition due to a germline variant has important implications for risk-reducing interventions, cancer screening, and germline testing for the affected patient and their close relatives [2,3] as well as for treatment decisions. Thus, germline genetic testing has become an integral part of the care of patients with BC and their families since *BRCA1* (OMIM #113705) and *BRCA2* (OMIM #600185) were identified in 1994 and 1995, respectively [4,5]. In fact, about 5% of all breast cancers bear a *BRCA1/2* germline pathogenic variant (PV) [6], being a tissue sensitive to PARP inhibitors [7], and thus treatment with this type of molecule is feasible.

In the case of BC, there are different biological types that show considerable tissue-heterogeneity and thus extensive research has taken place for subtyping breast cancer at the molecular and genetic level to determine the various clinical, pathological, and molecular factors for the selection of treatment modalities and to visualize the prognosis of the disease at the time of diagnostics [8,9].

Breast cancers are categorized into three main groups based on cellular markers: (i) positive for estrogen receptors (ERs) and/or progesterone receptors (PRs); (ii) positive for amplification of human epidermal growth factor receptor 2 (HER2) with or without ER and PR positivity; and (iii) triple-negative breast cancer (TNBC), defined by a lack of ER/PR expression and HER2 amplification [8,10,11]. Targeted therapy is well defined and available for categories (i) and (ii); however, due to the aggressiveness and variety of the TNBC biology, frequently, treatment needs to be tailored according to the patient [12,13,14].

TNBC accounts for 10–20% of invasive breast cancers [6] and the developing risk is highly associated with age, race, and genetics [15]. Particularly, it is most frequently associated with younger pre-menopausal women (<42 years old) [15,16].

The pathogenic variant is present in about 20 to 50% of cases with BC and strong family history; individuals with such inheritance have a 50–80% risk of developing breast cancer in their lifetime [17,18]. However, much light needs to be shed on the genetics behind the development of TNBC because of the high aggressiveness of this type of BC and the necessity to tailor the treatment according to the patient. Germline pathogenic variants in the *BRCA1* and *BRCA2* genes have been associated with TNBC, with 60–70% of carriers of pathogenic variants displaying a TNBC phenotype [19]. Additional studies have identified *BRCA1/2* pathogenic variants in up to 29% of patients of Ashkenazi Jewish ethnicity presenting with TNBC [20], 20% of those with TNBC diagnosed at a young age and/or with a family history of breast cancer, and 8–14% of those with TNBC unselected for family history [21,22].

On the other hand, increased knowledge about the human genome and advances in genomic technology have made it possible to simultaneously examine genes in exome/gene panels. In fact, recent advances in next-generation sequencing (NGS) technology have enabled simultaneous sequencing of multiple-gene panels [23], expanding the association of many cancers with different genes, although the results up to now have not filled the expectations, with a pathogenic variant detected conferring a much higher risk of developing cancer than those in the general population [24]. In this actual paradigm, in which the use of extensive multi-gene panels has become the standard for the identification of new genes and LPVs/PVs associated with BC, guidelines for genetic testing and strategies for cancer surveillance and prevention have been developed and incorporated into oncologic practice [25].

In this paper, we first examine and describe the findings in a large single-center study in South America (2155 cases), comparing the frequency and type of variants found in the subgroup of TNBC patients. Additionally, it was observed that *BRCA1/2* germline PVs might be associated with prolonged survival only if women were diagnosed with TNBC [26].

We also examine the incremental of the LPV/PV frequency in inherited susceptibility genes detected by a broad universal testing strategy of *BRCA1/2* analysis, compared to the use of the multiple-panel gene NGS strategy. This final analysis aims to evaluate the improvement in the percentage of detection of LPV/PV, as well as evaluate the cost-effectiveness that the massive use of multiple panels can have in countries, such as Argentina, where the conditions of economic inequality in the population are preponderant.

## 2. Materials and Methods

### 2.1. Study Subjects

In this study, we included 1900 individuals selected by familial (at least two relatives, one of 1st and one of 2nd degree) and/or personal history of BOC analyzed for *BRCA1* and *BRCA2* genes referred to the Laboratory of Genotyping at Centro de Educación Médica e Investigaciones Clínicas Norberto Quirno (CEMIC), a university hospital located in Buenos Aires, Argentina, between January 2016 and December 2020. The criteria for inclusion in the gene panel study, recruiting 255 cases, as to have BOC not clearly associated with *BRCA1/2* genes and thus the study was expanded to other related genes, or individuals with a personal and/or family history with BOC and related tumors, as selected by the clinician.

The inclusion criteria for the selection of the actionable genes included in the multi-gene panel were based on the recommendations in the Clinical Practice Guidelines in Oncology of the National Comprehensive Cancer Network guidelines (NCCN Guidelines^®^)-Genetic/Familial High-Risk Assessment: Breast, Ovarian and Pancreatic (Version 1.2020) [25].

### 2.2. Samples Preparation

Genomic DNA were extracted from blood samples with EDTA in the MagNA^®^ Pure LC instrument with a total DNA isolation kit I (Roche Diagnostics Argentina, Buenos Aires, Argentina), following the manufacturer’s instructions.

### 2.3. BRCA1/2 Testing

Sequence analysis of complete *BRCA1* and *BRCA2* was performed by next-generation sequencing (NGS) and large rearrangements by Multiple Ligation-dependent Probe Amplification assay (MLPA–MRC Holland, Amsterdam, Netherlands). The promoter region was not sequenced in NGS, and was only assayed by MLPA, according to the manufacturer’s instructions.

The NGS sequencing platform used to process patient samples was Illumina^®^ (San Diego, CA, USA). Assays were designed to ensure at least 200× total coverage/base and amplify the complete coding sequences of *BRCA1* and *BRCA2* genes, including 20 to 50 intronic adjacent bases to each exon. Sequences with low coverage were also analyzed by Sanger to guarantee the total coverage of genes. Sanger sequencing was also used for the confirmation of variants detected by NGS that have clinical relevance (Class 4: likely pathogenic; and class 5: pathogenic).

### 2.4. Multiple-Gene Panel Analysis

The samples were analyzed by complete exome sequencing (WES) with the Illumina^®^ platform with similar technic specifications as before (see *BRCA1/2* testing) and filtered for the following genes: *ATM, BRCA1, BRCA2, BRIP1, CDH1, CHEK2, CDKN2A, EPCAM, MUTYH, NF1, NBN, PALB2, PMS2, PTEN, RAD51C, RAD51D, STK11,* and *TP53*. The genes filtered and analyzed for each individual are according to the clinician order. As well as for *BRCA1/2* testing, sequences with low coverage were also analyzed by Sanger to guarantee the total coverage of the sequence and if it was the case, we confirmed the variants detected with clinical relevance (class 4: likely pathogenic; and class 5: pathogenic).

### 2.5. Large Rearrangements of BRCA1/2

Large rearrangements were measured by MLPA using SALSA MLPA Probemix P002-*BRCA1* and SALSA MLPA Probemix P045-*BRCA2*/*CHEK2* (MRC-Holland, Amsterdam, Netherlands) according to the manufacturer’s recommendations.

We confirmed the positive results with SALSA MLPA Probemix P087 and SALSA MLPA Probemix P077 for *BRCA1* and *BRCA2*, respectively.

### 2.6. Data Analysis

The variants were nominated following the HGVS nomenclature rules. The transcripts used included: *ATM:* NM_000051.3; *BRCA1*: NM_007294.3; *BRCA2*: NM_000059.3; *BRIP1*: NM_32043.3; *CDH1*: NM_004360.3; *CHEK2*: NM_007194.3; *CDKN2A*: NM_000077.4; *EPCAM*: NM_2354.3; *MUTYH*: NM_01128425.1; *NF1*: NM_ 000267.3; *NBN*: NM_002485.4; *PALB2*: NM_24675.3; *PMS2*: NM_000535.5; *PTEN*: NM_000314.6; *RAD51C*: NM_058216.3; *RAD51D*: NM_002878.3; *STK11*: NM_000455.4; and *TP53*: NM_000546.5.

For all NGS, *BRCA1/2* or multi-panel full sequencing studies, BAM files were analyzed in Alamut Software^®^. Through this procedure, we checked the adequate vertical and horizontal coverage of each gene requested for analysis. The presence of all the variants reported by the annotation file was verified and we extended the control for the detection of additional variants that were missed for some reason: low coverage of the region, low allelic frequency, or simply not listed because of an informatic error. This step is especially important to ensure the correct coverage and annotation of the variants for the final report. As previously mentioned, the presence of the pathogenic variants (class 4 and class 5) was confirmed by Sanger sequencing, as well as the regions with a vertical coverage with less than 20 reads/base.

The interpretation and clinical classification of the genetic variants were carried out according to the recommendations of the American College of Medical Genetics and Genomics (ACMG) [27]. In this sense, a five-tier system was used for the variants: class 1 (benign), class 2 (likely benign), class 3 (variant with uncertain clinical significance, VUS), class 4 (likely pathogenic), and class 5 (pathogenic). We used the following reference databases for the clinical significance report: ClinVar [28], LOVD3.0 [29], and UMD [30], as of December 2020. In silico analysis was conducted for missense variants to predict the functional compatibility for amino acid changes with the software Align-GVGD [31], SIFT [32], and Mutations Taster [33].

### 2.7. Statistical Analysis

GraphPad Prism (GraphPad Software, Inc., Jolla, CA, USA) was used for statistical studies. Statistical significance was determined by ANOVA followed by the Tukey test.

## 3. Results

In total, 2155 patients were analyzed, with 1900 patients analyzed for *BRCA1/2* genes and 255 patients studied for a panel of the following genes: *ATM, BRCA1, BRCA2, BRIP1, CDH1, CDKN2A, CHEK2, EPCAM, MUTYH, NF1, NBN, PALB2, PMS2, PTEN, RAD51C, RAD51D, STK11*, and *TP53*, as described in the materials and methods, including 2074 women, as summarized in Table 1.

The 2155 patients comprised 443 women with TNBC, 30 of which were studied with the panel of genes, resulting in two of them having a non-*BRCA* PV, as shown in Table 2. The mean age of diagnosis for the gene-panel group was 42.4 ± 10.7 years, significantly different from the TNBC group: 32.7 ± 9.7. The panels analyzed included two healthy women and one man, none of whom were a carrier of the LPV/PV detected.

The total LPV/PV of the 2155 samples sequenced was 98.13% for *BRCA1/2* and the rest (1.87%) for non-*BRCA1/2* variants.

In Table 2, the LPV/PV in non-*BRCA* genes are listed, showing that most of the variants are class 5 (PV) and only two are class 4 (LPV). Interestingly, PV c.1528dup in *CDH1* is novel to our knowledge.

Table 3 summarizes the results in 443 TNBC patients in both PV in *BRCA1/2* and non-*BRCA* genes, showing a clear preponderance for PV in *BRCA1*, less than half in *BRCA2*, and little presence in non-*BRCA* genes.

The full list of variants detected in the TNBC patients is listed in Table 4. Two cases have special characteristics: (a) AN500 has two PVs, as follows: a large rearrangement in *BRCA1* c.(-?_-232)_(80+1_81-1)del and a spliceogenic PV in *BRCA2* c.1909+1G>A. Her personal history begins with an ovary carcinoma at age of 42 years with retroperitoneal and lung metastasis, to follow at 46 years old with a TNBC; details regarding the family history are lacking, with only two relatives with cancer. Her father was diagnosed with colon cancer at 74 years old and her paternal grandmother died of uterus cancer at 80 years old; (b) AN519 is a 26-year-old woman, bearing three pathogenic variants coexisting in exon 5 of *BRCA1*, one of which is the recurrent variant of Spanish origin c.211del, p.(Arg71Glyfs*17) and the other two are c.187_188del, p.(Leu63Metfs*2), and c.191G>A, p.(Cys64Tyr) with a personal history of metachronic bilateral breast carcinoma, TN in right breast to follow at the age of 38 years with a ductal infiltrating carcinoma in the contralateral breast. The family history is very limited: paternal uncle has prostate cancer at 65 years and the paternal grandmother died of breast cancer at 87 years.

Regarding the patients with ovary cancer, they were in a vast part previously described [34], and no further relevant data needed to be incorporated.

It is interesting that the high rate of above 50% in LPV/PV detected in patients diagnosed with both cancers (breast and ovary, synchronic, or metachronic) is kept in those numbers since our results published in 2012 [35] with only 134 patients, maintained with 940 probands [36] and consolidated in this work with 2155 patients in which the patients with both cancers showed a detection rate of a PV in *BRCA1/2* of 51.8%

Regarding the male patients, as depicted in Table 5, with only one exception that was diagnosed with melanoma, the rest are healthy male carriers of PV in *BRCA1/*2.

Table 6 shows a summary of the tumors diagnosed in men that did not have LPV/PV in the NGS sequence for the *BRCA1/2* genes and the carriers of PV.

In the 1900 *BRCA1/2* sequences, there were 18 healthy women (Appendix A) and 12 male carriers of PV (Table 5). Appendix A lists the 199 women with PV with the spectrum of diagnosis in number of cases as follows: breast cancer (no-TNBC), 40 and 45; ovary cancer, 41 and 36; breast and ovary cancer, 8 and 10; and no cancer, 12 and 6, in *BRCA1* and *BRCA2*, respectively, and a single case of melanoma with the recurrent PV in *BRCA2* c.2808_2811del, p.(Ala938Profs*21) in a 37-year-old proband.

In Table 7, the recurrent variants (detected three or more times) in non-related probands in *BRCA1/2* among the total of 1900 patients are depicted. The three PVs of the Ashkenazi panel (c.68_68del and c.5266dup in *BRCA1* and c.5946del in *BRCA2*) were not detected in non-Ashkenazi patients and are not included in the recurrent pathogenic variants.

Since the men we received as probands do not have a pathogenic variant and are healthy (with one exception with a melanoma, see Table 5), we considered this to be a good sample of what we have in Argentina for male carriers of PV. The relatives detected from the proband’s families discussed in this work, all of them healthy, are listed in the Appendix A (not included in the 2155 subjects of the cohort).

## 4. Discussion

The goal of this work was to update the spectrum of LPV/PV in *BRCA1/2* and the first description of non-*BRCA* genes in Argentina associated with BOC, in particular in TNBC. The data is up to December 2020, based on 2155 new cases in women and men, with associated personal and/or family history, including the regional genetic variants launched in an early publication of ours [35]. In fact, in our actual reports of genetic variants, there are very few VUS present as a result of the more than 3000 fully sequenced patients deposited in the LOVD database, our platform for sharing data since 2016. In fact, we had many “VUS” highly frequent in our population and a high rate of them were coexistent with PV in *BRCA1/2* [37]; thus, the high population frequency and the coexistence with PV lowered the classification to benign variants. As a sharing-data laboratory deposited at LOVD, we are listed at “ar.lovd.org” [38]. From our Country Node of the Human Variome Project, we encourage others to deposit the variants and, as a result, laboratories from many provinces in the country are depositing their genetic variants [39].

Besides, in one of our previous publications [36], we reported in the conclusions a high percentage of novel variants (pathogenic or not) in *BRCA1/2*. This is very contrasting data with the present work in which doubling of the number of cases with no novel variants in *BRCA1/2* (only one novel variant in the CDH1 gene, c.1528dup). This is a result of the great amount of information in the last few years and the relevance of sharing data, which is key for the worldwide open genetic information to all scientists.

Table 1 shows a summary of the present study showing the *BRCA1/2* and non-*BRCA* PV in the cohort, in 2074 women with 67.6% of the mutations in *BRCA1*, the most frequent mutated gene in most series [19].

In response to the demand from oncologists and genetic counselors for the analysis of panels of genes, 255 panels were analyzed for the genes composed by: *ATM, BRCA1, BRCA2, BRIP1, CDH1, CDKN2A, CHEK2, EPCAM, MUTYH, NF1, NBN, PALB2, PMS2, PTEN, RAD51C, RAD51D, STK11*, and *TP53.* The results in Table 2 show the LPV/PV detected with 13 LPV/PV, with only 2 LPV (c.1169A>C and c.1427C>T) and the rest were PV. The novel variant in CDH1, c.1528dup, is a frameshift resulting in an expected truncated protein p.(Ala510Glyfs*27) in codon 537 and this is one codon reported as pathogenic (stop codon 536) at the LOVD data base (variant ID at LOVD: 00032560, 2004). This is finding supports the PV classification in concordance with the in silico analysis results.

A reiterate finding in the survival curve for TNBC exists, in which patients commonly show a sharp decrease in survival during the first 3 to 5 years after diagnosis. This type of breast cancer is biologically aggressive and many are potentially curable, reflecting their heterogeneity [40,41]. Thus, the eventual association with gene variants might be of great help in surveillance issues in at least two goals: screening for high-risk cases and treatment with specific drugs in the context of precision medicine for TNBC. For this reason, we focused on the TNBC patients in this cohort since it might help achieve both goals as well as identify regional variants in a zone of the world lacking large series analysis.

Most of the variants in TNBC (Table 3) have a PV in the *BRCA1* gene (70 patients) with a median age in years of 39.5 ± 9.5 and a range of 18 to 63 years, compared to the *BRCA2* PV with 34 probands with 42.5 ± 9.8 years (range 26–78); only one patient showed a PV in both genes. This is consistent with other publications reporting a greater rate of association with TNBC [19]. Only two non-*BRCA* gene variants were detected in this series of TNBC.

The Ashkenazi PV c.68_69del is present in 12 TNBC patients (vs. two patients in the non TNBC listed in Appendix A). This is a strong association for this variant as a predictor of a high risk in women bearing this variant. A similar situation is observed the recurrent variant in *BRCA1* c.211G>A, 7 times in TNBC.

Regarding the male patients, there are other publications concentrated on this matter describing tumors in men [42] that give an excellent description, concluding that “surveillance programs in men with *BRCA1* and *BRCA2* pathogenic variants should be tailored in light of these gene-specific cancer phenotype differences. These results may inform the design of prospective studies on cancer risks in male *BRCA1* and *BRCA2* pathogenic variant carriers”. As contributors in this publication, we take these conclusions since in the actual series, we did not recruit enough men to allow us to draw a summary, although it reinforces the silent manifestations in the men as carriers of PV in *BRCA1/2*. Table 6 shows that the age of diagnosis is beyond 50 years and the unique diagnosis, a melanoma, was at 62 years.

Table 7 shows the recurrent PV in *BRCA1/2,* in which 9 variants in total with a 2.74% coverage is a very low rate to be used for public health purposes, i.e., in a panel for lowering the cost for population screening. This confirms the lack of utility of the “Hispanic panel” [43] in our population as we already anticipated [36]. In conclusion, the recurrent mutations were not enough to anticipate a panel of PV to simplify the testing in the population unfortunately.

The Appendix A shows a list of 199 PV in non-TNBC detected in the 1900 patients analyzed and the age (range) in years for each cancer as follows: *BRCA1*: BrCa, 44.1 ± 8.8 (n = 41); OvCa: 52.9 ± 10.4 (n = 41); BOC, 45.9 ± 7.9 (n = 8) and No tumor: 38.8 ± 10.0 (n = 12) and for *BRCA2*: BrCa, 43.1 ± 10.8 (n = 44); OvCa: 56.8 ± 7.0 (n = 36); BOC, 45.5 ± 10.4 (n = 10) and No tumor: 50.5 ± 9.0 (n = 6). The ages showed no significant difference between both genes in BrCa and BOC, which is somewhat surprising, although there may be too few cases to be conclusive. The group with OvCa and with no tumor were significantly different (*p* < 0.001 and *p* < 0.01, respectively), with an older age for the *BRCA2* gene, consistent with the reported data. The age at diagnosis for BrCa and OvCa in both genes showed significantly older probands for OvCa, consistently described in the majority of previous publications.

In most of the publications related to *BRCA1/2* in the general population, there is a preponderance of women. For this reason, we thought it might be informative for the list of healthy men carriers of a pathogenic variant in *BRCA1/2* with a wide range of age, 19–71, in Appendix A. These men were not included in the cohort since all are relatives of the probands analyzed (only affected or healthy probands with a strong family history were included in the study). All the PV are highly penetrant; this is the reason men are considered “silent” carriers (not mute).

The best news is the precision medicine treatments associated with *BRCA1/2* LPV/PV in TNBC preferably; detection ensured by good-quality DNA sequencing supporting the necessity of a high level in genetic reports as the basis of the best success for treatments. This data is also of value in surveillance programming for the highest risk in TNBC cases in breast cancer screening and the lessons from the regional spectrum of genetic variants in *BRCA1/2* and expecting to be enlarged, or not, in the non-*BRCA* variants.

## 5. Conclusions

In conclusion, TNBC is associated with *BRCA1/2* at a higher rate than the rest of the breast cancer types and targeted therapy treatment needs to be frequently tailored according to the patient. For these reasons, genetic testing is highly indicated. The most aggressive TNBC calls for stronger surveillance of carriers with higher probabilities of developing this type of cancer and, also, since the diagnosis is usually at younger age, the detection of carriers helps to prevent cancer in these persons. The more prevalent LPV/PV in *BRCA1/2* genes (mostly in *BRCA1*) does not rule out the importance of panels of genes being tested.

## Figures and Tables

**Table 1 cancers-13-02711-t001:** Probands analyzed for *BRCA1/2* genes and the panel of genes.

Probands	Number of Probands (% of Total)
Total analyzed	2155
Women	2074
*BRCA1/2*	1900
Panel of genes	255
Likely pathogenic and Pathogenic variants	372 (17.2)
*BRCA1*	207 (9.6)
*BRCA2*	152 (7.0)
Non-*BRCA*	13 (0.6)

**Table 2 cancers-13-02711-t002:** Non-*BRCA* likely pathogenic/pathogenic variants in 255 women with a breast cancer diagnosis analyzed by a panel of genes.

ID	Age	Gene	Exon/Intron	HGVS c.	HGVS p.
AN620	55	*CDH1*	10	c.1528dup	Ala510Glyfs*27
AN621	57	*CDKN2A*	2	c.176T>G	Val59Gly
AN622	56	*CHEK2*	2	c.279G>A	Trp93*
AN609	29	*CHEK2*	3	c.349A>G	Arg117Gly
AN623	56	*CHEK2*	3	c.349A>G	Arg117Gly
AN624	56	*CHEK2*	8i	c.846+1G>C	
AN625 ^¥^	56	*CHEK2*	11	c.1169A>C	Tyr390Ser
AN626	50	*CHEK2*	11	c.1209_1233del	Tyr404Valfs*2
AN627 ^¥^	43	*CHEK2*	13	c.1427C>T	Thr476Met
AN628	43	*MUTYH*	12	c.1105del	Ala371Profs*23
AN629	46	*NBN*	6	c.657_661del	Lys219Asnfs*16
AN630	43	*RAD51D*	1	c.1A>G	Met1Val
AN610	35	*TP53*	7	c.742C>T	Arg248Trp

At diagnosis, mean ± SD years (range): 48.1 ± 9.1 (29–57). Individuals AN609 and AN610 are a TNBC patients. The panel is composed by the following genes: *ATM, BRCA1, BRCA2, BRIP1, CDKN2A, CDH1, CHEK2, EPCAM, MUTYH, NF1, NBN, PALB2, PMS2, PTEN, RAD51C, RAD51D, STK11*, and *TP53.*
^¥^ These are likely pathogenic variants; the rest are all pathogenic variants.

**Table 3 cancers-13-02711-t003:** Likely pathogenic/pathogenic variants in *BRCA1/2* and non-*BRCA* detected in triple-negative breast cancer cases.

TNBC Patients	Number
Total	443
Likely pathogenic/Pathogenic	107 (24.1%)
*BRCA1/2*	105 (23.7%)
PV in *BRCA1* (66.7%)	70
PV in *BRCA2* (32.4%)	34
PV in *BRCA1 & 2* (0.9%)	1
Non-*BRCA1/2*	2
LPV in *CHEK2*	1 (0.23%)
PV in *TP53*	1 (0.23%)

**Table 4 cancers-13-02711-t004:** Likely pathogenic/pathogenic variants detected in TNBC patients (n = 107).

ID	Age	Gene	Exon/Intron	HGVS c.	HGVS p.
AN500 ^§^	46	*BRCA1*	_2_2i	c.(-?_-232)_(80+1_81-1)del	
AN501	39	*BRCA1*	2	c.68_69del	Glu23Valfs*17
AN502	46	*BRCA1*	2	c.68_69del	Glu23Valfs*17
AN503	43	*BRCA1*	2	c.68_69del	Glu23Valfs*17
AN504	30	*BRCA1*	2	c.68_69del	Glu23Valfs*17
AN505	35	*BRCA1*	2	c.68_69del	Glu23Valfs*17
AN506	36	*BRCA1*	2	c.68_69del	Glu23Valfs*17
AN507	50	*BRCA1*	2	c.68_69del	Glu23Valfs*17
AN508	49	*BRCA1*	2	c.68_69del	Glu23Valfs*17
AN509	37	*BRCA1*	2	c.68_69del	Glu23Valfs*17
AN510	37	*BRCA1*	2	c.68_69del	Glu23Valfs*17
AN511	35	*BRCA1*	2	c.68_69del	Glu23Valfs*17
AN512	44	*BRCA1*	2	c.68_69del	Glu23Valfs*17
AN513	63	*BRCA1*	2i	c.81-1G>A	
AN514	37	*BRCA1*	3i	c.134+2T>C	
AN515	25	*BRCA1*	3i_24_	c.(134+1_135-1)_(*1383_?)del	
AN516	31	*BRCA1*	5	c.140G>T	Cys47Phe
AN517	31	*BRCA1*	5	c.181T>G	Cys61Gly
AN518	54	*BRCA1*	5	c.181T>G	Cys61Gly
AN519 ^¥^	26	*BRCA1*	5	c.187_188del	Leu63Metfs*2
AN520	46	*BRCA1*	5	c.187_191delinsATA	Leu63Ilefs*2
AN521	35	*BRCA1*	5	c.190_191ins19	Cys64*
AN522	47	*BRCA1*	5	c.190_191ins19	Cys64*
AN523	35	*BRCA1*	5	c.190T>C	Cys64Arg
AN519 ^¥^	26	*BRCA1*	5	c.191G>A	Cys64Tyr
AN525	50	*BRCA1*	5	c.211A>G	Arg71Gly
AN526	50	*BRCA1*	5	c.211A>G	Arg71Gly
AN527	40	*BRCA1*	5	c.211A>G	Arg71Gly
AN525	50	*BRCA1*	5	c.211A>G	Arg71Gly
AN528	46	*BRCA1*	5	c.211A>G	Arg71Gly
AN529	51	*BRCA1*	5	c.211A>G	Arg71Gly
AN530	48	*BRCA1*	5	c.211A>G	Arg71Gly
AN531	32	*BRCA1*	5	c.211A>G	Arg71Gly
AN519 ^¥^	26	*BRCA1*	5	c.211del	Arg71Glyfs*17
AN533	33	*BRCA1*	5i	c.213-11T>G	
AN534	25	*BRCA1*	7	c.427G>T	Glu143*
AN535	48	*BRCA1*	8	c.470_471del	Ser157*
AN536	47	*BRCA1*	11	c.1039_1040del	Leu347Valfs*2
AN537	26	*BRCA1*	11	c.1067del	Gln356Argfs*18
AN538	47	*BRCA1*	11	c.1504_1507del	Leu502Serfs*29
AN539	58	*BRCA1*	11	c.1687C>T	Gln563*
AN540	36	*BRCA1*	11	c.2296_2297del	Ser766*
AN541	41	*BRCA1*	11	c.3228_3229del	Gly1077Alafs*8
AN542	48	*BRCA1*	11	c.3331_3334del	Gln1111Asnfs*5
AN543	48	*BRCA1*	11	c.3627dup	Glu1210Argfs*9
AN546	35	*BRCA1*	11	c.3858_3861del	Ser1286Argfs*20
AN547	28	*BRCA1*	11	c.4042G>T	Gly1348*
AN548	38	*BRCA1*	11	c.4065_4068del	Asn1355Lysfs*10
AN549	48	*BRCA1*	11	c.4183C>T	Gln1395*
AN550	31	*BRCA1*	13	c.4201C>T	Gln1401*
AN551	38	*BRCA1*	13	c.4327C>T	Arg1443*
AN552	40	*BRCA1*	13i-14i	c.(4357+1_4358-1)_(4484+1_4485-1)del	
AN553	41	*BRCA1*	14	c.4392del	Ile1465*
AN554	18	*BRCA1*	14	c.4484G>T	Arg1495Met
AN555	38	*BRCA1*	15i	c.4675+2T>A	
AN556	40	*BRCA1*	15i-17i	c.(4675+1_4676-1)_(5074+1_5075-1)del	
AN557	38	*BRCA1*	15i-17i	c.(4675+1_4676-1)_(5074+1_5075-1)del	
AN558	38	*BRCA1*	16	c.4736_4739del	Pro1579Leufs*21
AN559	32	*BRCA1*	16	c.4964_4982del	Ser1655Tyrfs*16
AN560	55	*BRCA1*	16	c.4964_4982del	Ser1655Tyrfs*16
AN561	38	*BRCA1*	16i	c.4986+4A>C	
AN562	30	*BRCA1*	17	c.5030_5033del	Thr1677Ilefs*2
AN563	25	*BRCA1*	17	c.5030_5033del	Thr1677Ilefs*2
AN564	51	*BRCA1*	17	c.5030_5033del	Thr1677Ilefs*2
AN565	33	*BRCA1*	18	c.5095C>T	Arg1699Trp
AN566	31	*BRCA1*	18	c.5123C>A	Ala1708Glu
AN567	36	*BRCA1*	19i-20i	c.(5193+1_5194-1)_(5227+1_5278-1)del	
AN568	37	*BRCA1*	20	c.5266dup	Gln1756Profs*74
AN569	33	*BRCA1*	20	c.5266dup	Gln1756Profs*74
AN570	45	*BRCA1*	20	c.5266dup	Gln1756Profs*74
AN571	30	*BRCA1*	21	c.5282T>C	Phe1761Ser
AN572	37	*BRCA1*	23	c.5431C>T	Gln1811*
AN573	47	*BRCA1*	23	c.5445G>A	Trp1815*
AN574	54	*BRCA1*	23i_24_	c.(5467+1_5468-1)_(*1383_?)del	
AN575	38	*BRCA2*	2	c.51_52del	Arg18Leufs*12
AN576	38	*BRCA2*	2	c.51_52del	Arg18Leufs*12
AN577	44	*BRCA2*	3	c.156_157ins(Alu)	
AN578	61	*BRCA2*	3	c.214A>C	Asn72His
AN579	46	*BRCA2*	6i	c.516+3A>G	
AN580	46	*BRCA2*	10	c.1337T>A	Leu446*
AN500 ^§^	46	*BRCA2*	10i	c.1909+1G>A	
AN582	45	*BRCA2*	11	c.2657del	Asn886Metfs*9
AN583	36	*BRCA2*	11	c.2808_2811del	Ala938Profs*21
AN584	30	*BRCA2*	11	c.2808_2811del	Ala938Profs*21
AN585	40	*BRCA2*	11	c.2808_2811del	Ala938Profs*21
AN586	40	*BRCA2*	11	c.2808_2811del	Ala938Profs*21
AN587	32	*BRCA2*	11	c.2830A>T	Lys944*
AN544	43	*BRCA2*	11	c.3744_3747del	Ser1248Argfs*10
AN545	56	*BRCA2*	11	c.3847_3828del	Val1283Lysfs*2
AN588	41	*BRCA2*	11	c.4277del	Thr1426Asnfs*22
AN589	38	*BRCA2*	11	c.4928T>C	Val1643Ala
AN590	26	*BRCA2*	11	c.5351dup	Asn1784Lysfs*3
AN591	47	*BRCA2*	11	c.5682C>G	Tyr1894*
AN592	43	*BRCA2*	11	c.5946del	Ser1982Argfs*22
AN593	78	*BRCA2*	11	c.5946del	Ser1982Argfs*22
AN594	44	*BRCA2*	11	c.6024dup	Gln2009Alafs*9
AN595	41	*BRCA2*	11	c.6024dup	Gln2009Alafs*9
AN596	57	*BRCA2*	11	c.6024dup	Gln2009Alafs*9
AN597	33	*BRCA2*	11	c.6275_6276del	Leu2092Profs*7
AN598	40	*BRCA2*	11	c.6405_6409del	Asn2135Lysfs*3
AN599	28	*BRCA2*	11	c.6596del	Thr2199Ilefs*7
AN600	41	*BRCA2*	15	c.7480C>T	Arg2494*
AN601	42	*BRCA2*	18	c.7985C>T	Thr2662Met
AN602	38	*BRCA2*	19	c.8351G>A	Arg2784Gln
AN603	38	*BRCA2*	19i	c.8487+1G>A	
tAN604	39	*BRCA2*	21i	c.8754+4A>G	
AN605	45	*BRCA2*	21i	c.8755-1G>A	
AN606	57	*BRCA2*	22	c.8942A>G	Glu2981Gly
AN607	33	*BRCA2*	25	c.9481A>T	Lys3161*
Non-*BRCA1/2*					
AN609	29	*CHEK2*	3	c.349A>G	Arg117Gly
AN610	35	*TP53*	7	c.742C>T	Arg248Trp

^§^ Two pathogenic variants in *BRCA1* and *BRCA2* coexisting in the same patient AN500. ^¥^ Three pathogenic variants in exon 3 of *BRCA1* coexisting in the same patient AN519.

**Table 5 cancers-13-02711-t005:** Men with pathogenic variants in BRCA1/2; n = 12; age, in years ± SD (range): 48.8 ± 14.9 (19–76).

ID	Age	Tumor	Gene	Exon/Intron	HGVS c.	HGVS p.
MC134	43	No	*BRCA1*	2	c.68_69del	Glu23Valfs*17
MC162	53	No	*BRCA1*	2	c.68_69del	Glu23Valfs*17
MC166	43	No	*BRCA1*	2	c.68_69del	Glu23Valfs*17
MC167	31	No	*BRCA1*	20	c.5266dup	Gln1756Profs*74
MC186	44	No	*BRCA2*	2	c.51_52del	Arg18Leufs*12
MC140	59	No	*BRCA2*	10	c.1670T>G	Leu557*
MC181	56	No	*BRCA2*	11	c.2808_2811del	Ala938Profs*21
MC130	62	Melanoma	*BRCA2*	11	c.5796_5797del	His1932Glnfs*12
MC146	19	No	*BRCA2*	11	c.5946del	Ser1982Argfs*22
MC127	43	No	*BRCA2*	11	c.5946del	Ser1982Argfs*22
MC126	76	No	*BRCA2*	11	c.5946del	Ser1982Argfs*22
MC171	56	No	*BRCA2*	17	c.7857G>A	Trp2619*

**Table 6 cancers-13-02711-t006:** Summary of the results in male patients.

Male Patients	Number	Age, Years ± SD (Range)
Total	81	
Without LPV/PV detected		
With tumor	44	54.6 ± 12.2 (29–78)
BrCa	13	55.2 ± 13.4 (29–72)
BrCa/Melanoma	1	45/70
Pancreas Ca	7	52.4 ± 9.2 (38–65)
Prostate Ca	20	56.5 ± 9.7 (40–78)
Prostate Ca/Melanoma	2	40/38 and 57/20
Prostate Ca/BrCa	1	67/69
Healthy (*)	25	53.1 ± 10.6 (33–67)
With PV detected	12	
Healthy carriers (**)	11	48.8 ± 14.3 (19–76)
Melanoma	1	62

* 5 out of the 25 patients were tested by the panel of genes as described in the materials and methods. ** 8 out of the 12 patients bear a mutation in the *BRCA2* gene, the rest in *BRCA1* (see Table 5).

**Table 7 cancers-13-02711-t007:** Recurrent pathogenic variants in *BRCA1/2* detected in 1900 probands with personal and/or family history of breast/ovary cancer.

Pathogenic Variant Total Pathogenic Variants Detected = 359	Unrelated Probands (% of the Total Probands)
*BRCA1*	
c.211A>G-p.(Arg71Gly)- ^¥^ (7)	13 (0.68)
c.1360_1361del-p.(Ser454*)	3 (0.16)
c.4964_4982del-p.(Ser1655Tyrfs*16)- ^¥^ (2)	4 (0.21)
c.5030_5033del-p.(Thr1677Ilefs*2)- ^¥^ (2)	4 (0.21)
c.5123C>A-p.(Ala1708Glu)- ^¥^ (1)	3 (0.16)
*BRCA2*	
c.517G>T-p.(Gly173Cys)	3 (0.16)
c.1909+1G>A- ^¥^ (1)	4 (0.21)
c.2808_2811del-p.(Ala938Profs*21)- ^¥^ (4)	9 (0.47)
c.5351dup-p.(Asn1784Lysfs*3)- ^¥^ (1)	52 (2.74)
Total recurrents = 9 PV	
Total recurrents (52)/total PV (359) %	14.4%

Note: in a previous series, we describe two other recurrent pathogenic variants: c.181T>G-p.(Cys61Gly), detected in one proband in this cohort, and the c.6037A>T-p.(Lys2013*) never present in the 1900 probands [36]. ^¥^ These pathogenic variants (PVs) are present in TNBC; in parentheses, the number of cases bearing the PV.

## Data Availability

The data presented in this studio are openly available in Leiden Open Variation Database, reference number [29].

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
