# Peer review of "Study of the Genetic Variants in BRCA1/2 and Non-BRCA Genes in a Population-Based Cohort of 2155 Breast/Ovary Cancer Patients, Including 443 Triple-Negative Breast Cancer Patients, in Argentina"

_cancers, 2021, doi:10.3390/cancers13112711_

Round 1

Reviewer 1 Report

A very interesting original article about gene sequencing in hereditary breast and ovarian cancer. I have some queries:

The paper needs better formatting, as there are various types of characters with the sections.

Ethical Issues should not be reported in the text, but may be reported in the Institutional Review Board Statement.

In the statistical analysis subsection, the program used to calculate means, standard deviations, and other tests for statistical significance was not reported; also, report the maker and its location.

Page 2 line 55-57 "Breast cancers are categorized into three main groups based on cellular markers: (i) positive for estrogen receptors (ER) and/or progesterone receptors (PR); (ii) positive for amplification of human epidermal growth factor receptor 2 (HER2) with or without ER and PR positivity" this paragraph needs a reference, such as: doi: 10.1007/s40264-021-01071-1. 

the phrase "The best news are the treatments associated with BRCA1/2- 389
LPV/PV, recently approved. ", repeated in the abstract and in the conclusion, is, in my opinion, useless, as not fully developed within the text and may be removed.

Thank You

Reviewer 2 Report

Authors collected and analyzed likely pathogenic and pathogenic variants (LPV and PV) from 2155 Breast and Ovarian Cancer (BOC) patients, between January 2016 and December 2020. Data revealed that most of LPV and PV (98.13%) are located in BRCA1 and BRCA2 genes. Triple negative breast cancer (TNBC) patients showed higher rate of BRCA LPV or PV compared to non-TNBC patients. Among the non-BRCA variants, authors found a novel pathogenic variant in CDH1 (c.1528dup). Overall, the manuscript contributes to the improvement of the routine diagnosis of BOC with a comprehensive analysis of BOC genetic variants.  

Some details need to be fixed before publication:

  1. Table 1: Where it says "Pathogenic variants", it should say "Pathogenic and Likely Pathogenic variants"
  2. Table 2, figure footnote: Individuals AN625 and AN627 are TNBC patients. However, in table 3, AN609 and AN610 are the TNBC patients who carry non-BRCA variants.
    Also, AN610 appears duplicate in table 2 with different ages (44 and 39 yo respectively). In table 3, AN610 is 35 yo.
    In table 2, CHECK2 c.349A>G is carried by AN610 (44yo) while in table 3, same variant is carried by AN609 (29yo).

Questions:

  1. Does the BRCA sequencing covered the promoter regions? 
  2. From 81 sequenced men, only 12 men presented LP or PV, and 11 of them were healthy. Should men be included in the panel?
  3. Only 17.2% of tested patients showed LPV or PV, being 98.13% of them  in BRCA1 or BRCA2.
    1. What do you think we are missing about BOC genetics?
    2. Would it worth it to analyze BRCA transcription levels (RNAseq)
    3. Are those panels comprehensive enough?

Round 2

Reviewer 1 Report

The authors responded to all queries. The paper is publishable